# Do 'leaders' in change sound different from 'laggers'? The perceptual similarity of New Zealand English voices

Elena Sheard[1]*, Jen Hay[1,2], Joshua Wilson Black[1], Lynn Clark[1,2]

1 New Zealand Institute of Language, Brain and Behaviour, University of Canterbury, Christchurch, New Zealand, 2 Department of Linguistics, University of Canterbury, Christchurch, New Zealand

* elena.sheard@canterbury.ac.nz

## Abstract

Work on covariation in New Zealand English has revealed groups of speakers characterised by their back vowel spaces and status as 'leaders' or 'laggers' across a set of ongoing vowel changes. We investigate whether listeners hear speakers from different groups as perceptually distinct. We conduct a perception task in which New Zealanders rate the similarity of pairs of speakers. We use the results to create a two-dimensional perceptual similarity space by means of Multi-Dimensional Scaling, and test if speakers are organised within this space according to their back vowels, leader-lagger status, speed, or mean pitch. Results indicate higher pitched and faster speakers are perceptually distinct from lower pitched and slower speakers. Leaders are perceptually distinct from laggers if they are not markedly higher pitched. A Generalised Additive Mixed Model fit to the trial-by-trial ratings shows order effects, revealing that perception of similarity is not symmetrical. They also support the perceptual relevance of speaker speed, pitch and leader-lagger status.

## 1. Introduction

What information do listeners use to make social judgements about voices? The conventional starting point for perceptual research within the variationist paradigm is the sociolinguistic variable. Considerable work has shown how sociolinguistic variables vary across different speakers in production, and it is assumed that listeners can also interpret this variation as socially meaningful. Indeed, we have substantial evidence for the potential of different (combinations of) variants to affect listener evaluations of speaker macro- or micro-social characteristics [e.g., 1,2–5], and of (purported) social information about a speaker to influence listener categorisation of linguistic variables [e.g., 6–8].

However, researchers are often starting with variables known to (co)vary in production but with unknown perceptual relevance. This can pose significant methodological questions. Is it reasonable to assume they are even perceptible to listeners

**Data availability statement:** All markdown and anonymised data files are available from the public github repository (https://github.com/nzilbb/qb-pairwise-public).

**Funding:** This research was supported by a Royal Society of New Zealand Marsden Research Grant (21-UOC-107) awarded to Kevin Watson, Jen Hay and Lynn Clark. The funders had no role in study design, data collection and analysis, decision to publish, or preparation of the manuscript.

**Competing interests:** The authors have declared that no competing interests exist.

in spontaneous speech? Which social characteristics might they be associated with, and how can we test this association in perception? Rather than assuming answers to these questions, an alternative approach is to start with the speaker and work bottom-up. By focusing on who listeners perceive to sound similar and different to one another *first*, and the (socio)linguistic variables or social characteristics that contribute to the perceptual differentiation of speakers *second*, it is possible to gain direct insight into listener perception without having to make prior assumptions about the variables in question. Here, we start with speakers of New Zealand English (NZE) and ask what structural relationships emerge among these speakers when listeners assess their similarity based on their spontaneous speech. We then further examine the degree to which our variables of particular interest, covarying vowel patterns in NZE, are relevant to this structure.

The analysis was carried out using the R Programming Language [9]. All code and anonymised data to reproduce the findings is publicly available in a GitHub repository (https://github.com/nzilbb/qb-pairwise-public). The supplementary materials also contain the preregistration for the experiment and additional methodological and analytical details. However, readers do not need to refer to any external content to follow the manuscript and we specify which elements of the reported analysis differ from the preregistration.

## 2. Connecting the production of covarying New Zealand English monophthongs to their perception

Sociolinguistic research in the 21st century has shown increasing interest in how linguistic variables pattern together at both the individual and community level [e.g., 10–12]. There is now growing evidence that linguistic features do not work in isolation from each other but can exist within systematic patterns of covariation and coherence. Such patterns may be more socially motivated (e.g., shifts away from dialectal variants) or shaped by linguistic pressures (e.g., vowel chain-shifts). Brand et al. [13] is illustrative of a shift towards multivariate analyses, focusing on a set of 10 NZE monophthongs (FLEECE, KIT, DRESS, TRAP, START, STRUT, LOT, NURSE, THOUGHT, GOOSE) in data from the Origins of New Zealand English (ONZE) corpus [14]. Brand et al. [13] investigate the degree to which a given speaker's realisation of a single vowel is predictive of their realisations of other vowels, relative to the population once known physiological and socio-demographic sources of variation are controlled for. Their analysis revealed the existence of structured, systematic, patterns in vowel realisations, with no monophthong produced independently of all other vowels in the set [13].

Brand et al. [13] implemented a novel statistical methodology to track potential vowel systems or clusters which had the following pipeline:

(1) Generalised Additive Mixed Models (GAMMs) are fit to F1 and F2 measures for each monophthong

   ◦ Each GAMM includes relevant fixed effects (e.g., speech rate, gender, age) and random effects (e.g., word, speaker)

(2) Speaker random intercepts are extracted from each GAMM [cf. 15]

(3) The extracted speaker random intercept values for each variable (F1/F2 for each monophthong) are used as input for a Principal Component Analysis (PCA)

PCA is a multivariate dimension-reduction technique that reduces many variables (i.e., F1/F2 speaker intercepts for 10 vowels) to fewer Principal Components (PCs), onto which multiple, covarying vowels are loaded [see also 16]. PCA identified three main PCs, or distinct clusters of covarying vowels, in the ONZE data [13]. One cluster related to the back vowels THOUGHT, START and STRUT (the *back-vowel configuration*). Individual speakers with lower and fronter realisations of THOUGHT have backer realisations of START and STRUT, and vice versa. The second cluster related to sound change, with individual speakers consistently "leading" or "lagging" in the ongoing changes for KIT, FLEECE, DRESS, TRAP, NURSE and LOT (the *leader-lagger continuum*). The final cluster captured two pairwise relationships: START and LOT, and DRESS and GOOSE.

Hurring et al. [17] replicated and extended Brand et al. [13], analysing covariation of the same monophthongs with a different corpus of contemporary NZE, the QuakeBox [18,19]. The original QuakeBox project (QB1) collected high quality audio and video recordings of earthquake stories in multiple locations across Christchurch, New Zealand in 2011–2012. The QuakeBox 2 (QB2) project then re-recorded stories from a subset of the QB1 speakers in 2019–2020. Hurring et al. [17] applied the GAMMs-to-PCA methodology developed by Brand et al. [13] to the QB data from each time point, resulting in two PCs for QB1 and QB2. The first two vowel clusters in Brand et al. [13] (the back vowel configuration and leader-lagger continuum) were identified in both the QB1 and QB2 principal components. TRAP, DRESS, FLEECE, KIT, and NURSE are consistently loaded onto the leader-lagger continuum, START, THOUGHT, and STRUT onto the back-vowel configuration (we note GOOSE and STRUT F1 in Hurring et al. [17] are also loaded onto the former and LOT is loaded onto the latter). Moreover, both vowel clusters in the QuakeBox data have remained stable over time.

The results of Hurring et al. [17] uphold the results of the original ONZE analysis and highlight the persistence of covarying NZE vowel patterns on both the collective and individual level. The same speakers maintain their position in the leader-lagger continuum over time, even when the production of individual vowels may be changing in the community. The combined results of Hurring et al. [17] and Brand et al. [13], then, provide evidence for NZE monophthongs working together as part of a complex system within which speakers can be leaders of structurally unrelated changes (although some changes in the leader-lagger continuum such as the Short Front Vowels (KIT, DRESS and TRAP) are arguably, and likely, structurally related, there are not such clear structural explanations for their relationships to the other vowels loaded onto the same Principal Component). Brand et al. [13] suggest that the patterns within this system may reflect clusters of speakers with shared social characteristics, and/or subsystems of sounds that carry shared social meaning. Here, we take a first step towards connecting the spontaneous production of covarying NZE monophthongs to their perception by NZE listeners.

## 2.1. Analysing sociolinguistic perception

How do we approach listener perception within the variationist paradigm? Perceptual studies on how different (combinations of) variants affect listener evaluations of speakers tend to employ the Matched Guise Technique [20] or a verbal guise format [e.g., 21–23]. Listeners hear different "guises" of the same speakers containing different (frequencies of) variants of the variables in question in the former, and excerpts from different speakers in the latter. In both formats, listeners either rate speakers along a scale [e.g., 4,23–28], or make a categorical forced choice [e.g., 2,3,29] for prespecified social characteristics. Listeners can assess speakers in relation to macro-social characteristics such as age, ethnicity or race [e.g., 2,21,24,27], or more localised speaker attributes, styles and social personae [e.g., 3,30–32].

While some research has explored attitudes towards NZE accents relative to other English varieties [e.g., 33,34], very little work has looked at sociolinguistic evaluations of variables within NZE. Szakay [21] has shown that voice quality and speech rhythm are used by listeners in tasks involving perception of ethnicity. Bayard and Bartlett [35] have demonstrated a perceptual association between rhoticity and region. Gordon's [36] work eliciting responses to three NZE voices shows

that there are variables that listeners hear as socially distinct but does not reveal which ones. Perceptual dialectology work, in which participants label maps of NZ, shows consistent use of labels evoking social class, suggesting a general orientation toward a relationship between accent and social class [37]. Some analyses have shown that listeners use social information to adjust how they listen to vowels [e.g., 1,38], which suggests a relationship between these variables and social evaluation but does not explicitly demonstrate it. In sum, despite the extensive sociolinguistic work on the production of NZE, we have very little clear evidence about the social judgement or perception of specific sociolinguistic variables, either in isolation or combination.

The leader-lagger continuum and back-vowel configuration present, therefore, a methodological challenge to investigating the social meaning(s) they may carry for listeners. Both guise techniques require researchers to specify relevant social characteristics, however we do not yet know whether listeners perceive the leader-lagger continuum and back-vowel configuration at all. One alternative approach is to start with listener perceptions of speaker similarity and then test if listeners differentiate between speakers based on specific social characteristics or linguistic features, such as covarying vowel patterns. For example, work on the perception of regional dialects in the United States has used both listener ratings of the likelihood of pairs of speakers coming from the same region [39] and free classification groupings of speakers listeners perceive to be from the same regions [40] to show speakers are perceptually differentiated by both their region and accent markedness. Explorations of perceived voice similarity in psychology and forensic linguistics have also explored the acoustic correlates that drive judgements of perceived similarity, producing consistent evidence for the role of fundamental frequency, laryngeal differences and formant values in differentiating between speakers perceptually [e.g., 41–43]. There are few investigations of sociolinguistic perception that seek to first differentiate between speakers based on their perceived similarity, and then test whether a given variable is relevant to how they are differentiated [though see 44].

In this paper, we start with exploring the perceived similarity of NZE speakers to create a general structure of listener perception and ask whether the leader-lagger continuum or the back-vowel configuration differentiate between these speakers within this structure. If they do, we then have a sound foundation for future explorations of the potential connections between the vowel clusters and associated social meanings.

## 2.2. Sociolinguistic perception of variables in context

Our experiment uses spontaneously produced speech. As noted by Campbell-Kibler [45], sociolinguistic perceptual work tends to employ controlled audio stimuli [though see 21,24–26,30,45,46]. But, as the reanalysis of the perceptual data in Villarreal [25] by Villarreal and Grama [47] highlights, it is possible for researchers' assumptions about variables' relative perceptual importance to listeners, which must be made in techniques such as the Matched Guise Technique, to result in the variables which most influence listener evaluations in spontaneous speech being overlooked.

It is also well-established that paralinguistic variables inherent to speech, and particularly variable in spontaneous speech, such as voice quality, speech timing and pitch, are perceived by listeners and subject to social evaluations [e.g., 48]. For example, there is longstanding evidence for features of speech timing influencing perceptions of dialectal differences, including the folk linguistic concept of 'drawling' in the Southern United States [49] and Wells' [50] impressionistic claim that urban speakers speak faster than rural speakers [see discussion in 51]. Perceptual research has also shown listeners can not only distinguish between fast and slow speech and lower and higher pitched voices, but can associate different speech timings and voice pitches with different ages [e.g., 52], genders [e.g., 53–55], ethnicities [e.g., 56], and personality attributes [e.g., 57,58]. These associations can, however, be mediated by language familiarity, listener native language and the language used by the speaker [e.g., 59,60]. Sociolinguistic perceptual work has also produced evidence that pitch can interact with other variables in affecting the perceived masculinity and/or sexuality of cisgender men [e.g., 5].

It is possible, therefore, that the perceptual relevance of a given sociolinguistic variable(s) depends on what other features are also present, or that their social significance will be mediated or dwarfed by other features. However,

outside of investigations of perceptions of male masculinity and sexuality [e.g., 4,5 both consider pitch as a variable], the sociolinguistic perceptual studies that have used spontaneous speech tend to not examine whether paralinguistic features also contribute to listener evaluations or interact with their variables of interest [though see 2,29]. We investigate not only whether the covarying vowel patterns are relevant to listener perception of speakers, but whether speaker articulation rate and pitch also differentiate between speakers within the same structure of listener perception.

## 2.3. Research questions

We conduct a pairwise similarity rating task to address two linked research questions. Our primary research question, building on the literature above, is:

*Research question 1*: **What structural relationships emerge among speakers when listeners evaluate them based on their spontaneous speech?** That is, which speakers do listeners perceive to sound similar and different to one another within a multi-dimensional perceptual space, and are speakers that listeners perceive to sound different to one another differentiated by their covarying vowel patterns, their speed, and/or their pitch? Our preregistration predicts that one or both covarying vowel patterns will link to perceived speaker similarity, reading: "We want to identify the groups of speakers that listeners think sound similar to each other, and then compare these groups to the patterns of speakers identified in Hurring et al.(In prep). We hypothesise that the speakers that listeners perceive as sounding similar in the experiments of this study will align with one or more of the vowel patterns associated with different Principal Components in Hurring et al.(In prep). If they do not, this will imply that factors other than vowel formants are being used to perceptually group speakers, and we will conduct exploratory analysis to explore what these factors may be." We note that Hurring et al. [17] is the paper we refer to in our preregistration.

We also explore a secondary research question to understand the patterns in our participants' trial-by-trial responses, and what these patterns can tell us about the results of our preregistered task. Perceptual studies have demonstrated that listeners are more likely to notice and attend to (changes in) certain linguistic features than others (i.e., some features are more perceptually salient than others) [61–63]. Work on the sociolinguistic assessment of linguistic variants also points to the variable salience of different features [e.g., 64] and suggests that non-standard or innovative variants may affect listener perception more than standard or conservative variants (i.e., some variants are more socio-linguistically salient than others) [65–67]. Thus, it may be that pitch, articulation rate or vowels are not only inherently salient to different degrees, but their sociolinguistic or perceptual salience might depend on the extent to which a speaker favours certain variants of one variable or another.

There is also evidence that sociolinguistic judgements can be made early and be relatively 'bullet-proof' once made; listeners can be relatively immune to variables that run counter to their initial judgements [e.g., 31,66,68]. If one variable is particularly salient in a speaker's recording, and occurs early, it may be the dominant influence in listeners' responses. Moreover, work on perceived similarity shows that this is not always symmetric. As Tversky (1977) outlines, similarity can depend on what is the 'subject' and what is the 'referent', with the features of the subject being more heavily weighted than the features of the referent. Hodgetts and Hahn [69] also demonstrate asymmetry in a non-verbal implicit measure of similarity, arguing that similarity is influenced by how complex it is to transform one object into an another. Regardless of the mechanism, asymmetry may be even more acute when the stimuli are presented auditorily, as the listener must assess the first speaker along various dimensions before encountering the second speaker.

An advantage of the Multi-dimensional Scaling approach we take to answer research question 1 is that – in our counterbalanced design – we will be able to observe the structure that emerges despite any trial-by-trial order effects. However, we were also interested in explicitly exploring these effects, to see what they could teach us about how listeners were completing our task. Our secondary research question, then, is:

*Research question 2*: **What factors influence individual pairwise similarity ratings?** Does looking at the trial-by-trial response patterns reveal different information than the MDS in terms of how the speakers are perceived? In particular, does the order in which voices are presented affect the ratings given? This question is exploratory and was not preregistered.

## 3. Methodology: online pairwise similarity task

### 3.1. Stimuli

The experiment uses data from the same QB1 corpus data analysed in Hurring et al. [17]. The specific audio stimuli used in the experiments come from the QuakeBox recordings of 38 women aged 46–55 who met two conditions. First, that they were included in the PCA analysis in Hurring et al. [17]. Second, that they had consented to have the audio of their recording shared publicly [18]. We selected stimuli from a specific gender and age group in the QuakeBox corpus to reduce the risk of participants assessing speaker similarity along these social factors. All 38 women were Pākehā (New Zealanders of European background); 33 had grown up in the South Island, predominantly in the North Canterbury region (27). As there was only one Māori woman in this age group who met the two conditions, we chose to not include her in the stimuli. Stimuli were selected based on length (maximum 10 seconds) and the presence of the monophthongs analysed in Brand et al. [13] and Hurring et al. [17] (at least 5 of the 10 had to be present).

We also considered content when selecting the stimuli. Consistent with the design of the QB corpus, many QB participants' stories have upsetting aspects, and we did not want to continually expose our participants to such content, especially as some of them may have experienced the earthquakes themselves. We consequently focused on more positive parts of the recordings, like the sense of community or amusing aspects of their earthquake experiences. Where this was not possible, we ensured the stimuli topics were not explicitly negative (i.e., while some stimuli discuss damage to property, none talk about death or traumatic personal events). We also ensured that the clips did not contain information that could give an indication as to the speakers' social backgrounds (e.g., occupation, specific Christchurch suburbs, schools attended etc.).

### 3.2. Experiment

We recruited participants online between December 2023 and February 2024 using targeted social media ads. Participants were offered a $10 e-voucher. Participants had to be over the age of 18, be a speaker of NZE, and have lived in NZ since at least the age of 7. All participants provided written consent by completing an online form prior to being directed to the experiment, and they could withdraw at any time before submitting their responses. Upon completing the experiment, participants completed a background questionnaire. This experiment was reviewed and approved by the Human Research Ethics Committee at the University of Canterbury (2023/60/LR-PS).

Following the approved protocol, we ran the experiment online using a JavaScript application developed by Chan [70] and adapted for the current study. Each participant listened to a subset of the possible 703 combinations of the 38 stimuli, because listening to all possible combinations would take multiple hours. Longer experiment times can reduce participant engagement [71] and lead to more unreliable responses [e.g., 72,73]. As such, each participant listened to two blocks of 19 stimulus pairs. In each block, each speaker was heard once. We used a semi-random sampling procedure that distributed the possible combinations of stimuli pairs as evenly as possible across the stimuli subsets participants heard (see supplementary materials Section 4 for details). In effect, for approximately every 37 participants all 703 stimuli pairs are listened to once. We note that this is how the stimuli were distributed in building the experiment, but participant drop out meant that this did not translate exactly in practice (see Section 5.5. for further discussion). The order the audio stimuli were presented to individual listeners was randomised, as was the order of stimuli within each pair.

We tried to steer participants toward social judgements, rather than superficial judgements based on acoustic properties, or aspects of what was said. Participants were thus given the following instructions:

"Sometimes people sound similar because they are friends, have similar occupations or personalities, or grew up in the same area.

In this task, we want to know which speakers of New Zealand English you think sound similar to each other, and which speakers you think sound different to one another.

In this task you will listen to 40 pairs of audio clips from different New Zealand women talking about their experiences of the Christchurch earthquakes as part of the QuakeBox project. We would like you to rate how similar the women in each pair sound to you. We are interested in who you think sound similar based on the way they talk, rather than the things they say."

Participants listened to each pair of speakers and rated how similar they thought each pair sounded [cf. 74]. Participants were asked to rate each pair on a scale from 'not similar' (0) to 'similar' (1) using a slider interface; they did not see any numeric values on the scale. Participants could not progress to the next stimuli until both stimuli had been played in their entirety, and they had clicked on the slider. 140 listeners completed the task, which we note is fewer than our pre-registered goal of 180–200. We stopped data collection and started analysis once it was clear participant recruitment had slowed down, and a significant delay would be needed to recruit the final 40 participants. We removed outliers (n = 7) using a method diverging from our preregistration as described in the supplementary materials (Section 7) and report an analysis of 133 participants. The supplementary materials also include an analysis which follows the preregistered filtering method (Section 8).

## 4. Analysis and results

### 4.1. Creating a two-dimensional perceptual space of speaker similarity by means of multidimensional scaling analysis

In this section we conduct our preregistered analysis, applying multidimensional scaling (MDS) to the similarity ratings from the pairwise ratings task. MDS is a data-reduction technique that represents measurements of (dis)similarity among pairs of objects "as distances between points of a low-dimensional multi-dimensional space" [75]. The multi-dimensional space is a (pseudo-)perceptual space which, theoretically, corresponds to the cue(s) driving the perceived similarity of the objects. It is, therefore, ideal for analysing pairwise similarity ratings and is commonly used to quantify perceptual similarity of a range of objects, including speech. While MDS has received some application in sociolinguistic perceptual work [e.g., 40,44,76–78], it has primarily been applied in phonetic, psychological and forensic analyses of perceived voice and speaker similarity [e.g., 41,79,80]. A recurring finding is that fundamental frequency (F0) correlates with perceived similarity of speakers [e.g., 43,81–83]. Some studies have also found evidence for an influence of F1 and F2 measurements [e.g., 43,82,83]. However, no MDS analyses have explicitly considered the role of covarying vowel patterns, and few have considered audio stimuli with variable speech rates.

Following the procedure detailed in the supplementary materials (see Section 5.1), we apply a non-metric MDS method (M-splines) to the results of the experiment using the smacof R package [84]. We scaled the pairwise similarity ratings per participant and then took the mean rating for each stimuli pair across all participants to create a 38 x 38 similarity matrix. As MDS requires all numbers in the input matrix to be above 0, the individual scaled ratings were all brought above 0 by adding the minimum score to all ratings before calculating the mean. We then we applied MDS to a dissimilarity matrix derived from the 38 x 38 similarity matrix using the 'reverse' option from the *sim2diss* function [84].

MDS requires the researcher to specify the number of dimensions of the multi-dimensional space *a priori*. A conventional approach to determining the number of dimensions to specify is to rely on a rule-of-thumb cut off for 'stress', a measure of the fit of the MDS where a lower stress value corresponds to a better fitting analysis (i.e., how well the analysis explains the underlying structure). However, as stress will always decrease as the number of dimensions increases, we do not consider

relying on a single stress value to be best practice [see 85]. We additionally apply two permutation-based tests to inform the choice of dimensions. The first test is a novel method available via the *mds_test* function in the nzilbb.vowels R package [16]. The function implements a permutation and bootstrapping procedure to compare the distribution of stress reduction as the number of dimensions increases to the distribution of stress reduction we would expect if there were no structure in the data. The results of this procedure indicated that adding in a third dimension would not reduce stress more than we would expect by chance. The second test is an informal significance test [see 85] in which we use the *permtest* function from the *smacof* package to calculate stress for two-dimensional MDS analysis applied to 500 permuted iterations of the dissimilarity matrix [84]. The stress value for the unaltered matrix is below 95% of the permuted stress values, which we treat as informally equivalent to a *p*-value <0.05. Together, the results from the two tests support a two-dimensional MDS.

Following Mair et al. [85] we report the two-dimensional MDS with the lowest stress value (Stress-1 = 0.31) from 100 random starts. Fig 1 maps the coordinates from the two specified dimensions in our final MDS analysis for the 38 stimuli: Dimension 1 (D1) scores are on the horizontal axis, and Dimension 2 (D2) scores are on the vertical axis (Scree, shepard and bubble plots are also available in Section 5.1 of the Supplementary Materials). The closer two individuals are to each other in this space, the more similar they are perceived to sound. The further two individuals are from each other, the less similar they are perceived to sound.

### 4.1.1. Predicting perceptual dimensions from MDS analysis with regression trees and random forests.
MDS can reduce similarity data to a smaller number of dimensions, but the technique cannot provide insight into what each dimension represents or the cues with which they correlate. To investigate whether our variables of interest correspond to either D1 and D2, we fit two regression trees with D1 and D2 as the dependent variables. We also implemented random forests to evaluate the importance of each predictor in predicting each dimension. As outlined in detail in the supplementary materials (Section 5.2) we fit the regression trees and random forests using the parsnip package [86] in R, with the engine set to *rpart* for the former [87] and *ranger* [88] for the latter.

The independent variables in both regression trees were:

- Stimulus articulation rate

- Stimulus mean pitch

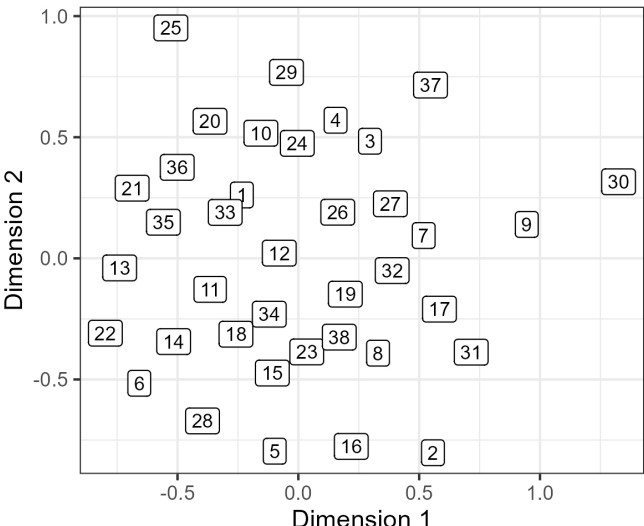

**Fig 1. Output of MDS mapped to two dimensions.**

- The speaker's position on the leader-lagger continuum (i.e., How much they are "leading" or "lagging" in the changes for FLEECE, DRESS, KIT, TRAP, NURSE, STRUT, GOOSE, based on their full QB1 monologue – see Hurring et al. [17])

- The speaker's position in the back-vowel configuration (i.e., the relationship between their START, THOUGHT and LOT vowels, based on their full QB1 monologue – see Hurring et al. [17]).

We preregistered a series of pairwise correlations between the perceptual dimensions and these factors, but moving to a regression tree is preferable as it allows us to examine how the factors might work together, rather than considering them independently. We report the preregistered correlations in the supplementary materials (Section 6).

The mean pitch measurements were extracted manually from the stimuli in Praat [89] using the default pitch range settings and the cross-correlation method. We have quantified speaker 'articulation rate' as the total number of canonical syllables they produced in their stimulus, divided by their total phonation time. Total phonation time was calculated based on the forced-alignment of participants' QB monologues in LaBB-CAT [90] and includes corrections, incomplete productions, the filled pauses *um* and *uh/ah*, and inter-word pauses less than 50 milliseconds in length. The number of canonical syllables is based on the CELEX dictionary used by LaBB-CAT in forced alignment. Speakers' positions in the leader-lagger continuum and back-vowel configuration are represented by their QB1 Principal Components loadings from Hurring et al. [17], which quantifies speaker position for these variables based on the monophthongs produced by the speaker across their entire monologue. All variables are scaled across the 38 speakers.

Fig 2A displays the regression tree predicting Dimension 1, while Fig 2B shows the regression tree predicting Dimension 2. Regression trees split the data into smaller groups called "nodes," then fit a model with the independent variables to each node. The tree then generates an if-else rule at each node based on the most important predictor, which further divides the data into subsequent nodes. This process continues until certain stopping criteria are fulfilled. Each node displays the estimated value of the dependent variable, along with the proportion and number of observations contained within that node.

The most important predictor of Dimension 1 is a speaker's mean pitch, with higher pitched speakers estimated to have a lower D1 score. Within the lower pitched speakers, there is a role of a speaker's position in the leader-lagger continuum, with laggers in this cluster of vowel changes (those with lower leader-lagger scores) estimated to have a lower D1 score, and leaders estimated to have a higher D1 score. As such, speakers with a higher D1 score are more likely to be lower pitched speakers who are leaders in the leader-leader continuum. The most important predictor of D2 is articulation rate, with slow speakers estimated to have a lower D2 score. Mean pitch also plays a role for the faster speakers, where fast, higher pitch speakers are estimated to have a higher D2 score.

The results of the random forest procedures uphold the relative importance of pitch, speed and the leader-lagger continuum in predicting Dimensions 1 and 2. Specifically, mean pitch, followed by the leader-lagger continuum emerge as the most important predictors of D1. Articulation rate, followed by mean pitch emerge as the most important predictors of D2. The random forest procedure also points to the back-vowel configuration as having a positive predictive effect on D2, albeit to a much lesser extent than speed and mean pitch (please refer to Section 5.2.2 of the supplementary materials for our discussion on this result). The results of the regression trees and random forests therefore point to these variables contributing to the perceived similarity of these speakers.

Fig 2A and Fig 2B also map the cutoff values in the regression trees onto the D1 and D2 coordinates in Fig 1. Fig 2A highlights the high-pitch speakers in red and the lower-pitch laggers and leaders (with a leader-lagger score below/above −0.26) in purple and yellow, respectively. Fig 2B highlights slow speakers (this time with an articulation rate below −0.27) in blue. Two groups of slower and lower pitch speakers are concentrated in the bottom right and left of the space. Fig 2B also reflects the perceptual importance of pitch, with the fast high pitch speakers concentrated in the top left of the space (with a mean pitch above 0.31) in red; we also see most (4/5) slow and higher pitched speakers concentrated in the left of the space. The fast, low-pitched speakers are concentrated in the middle in orange.

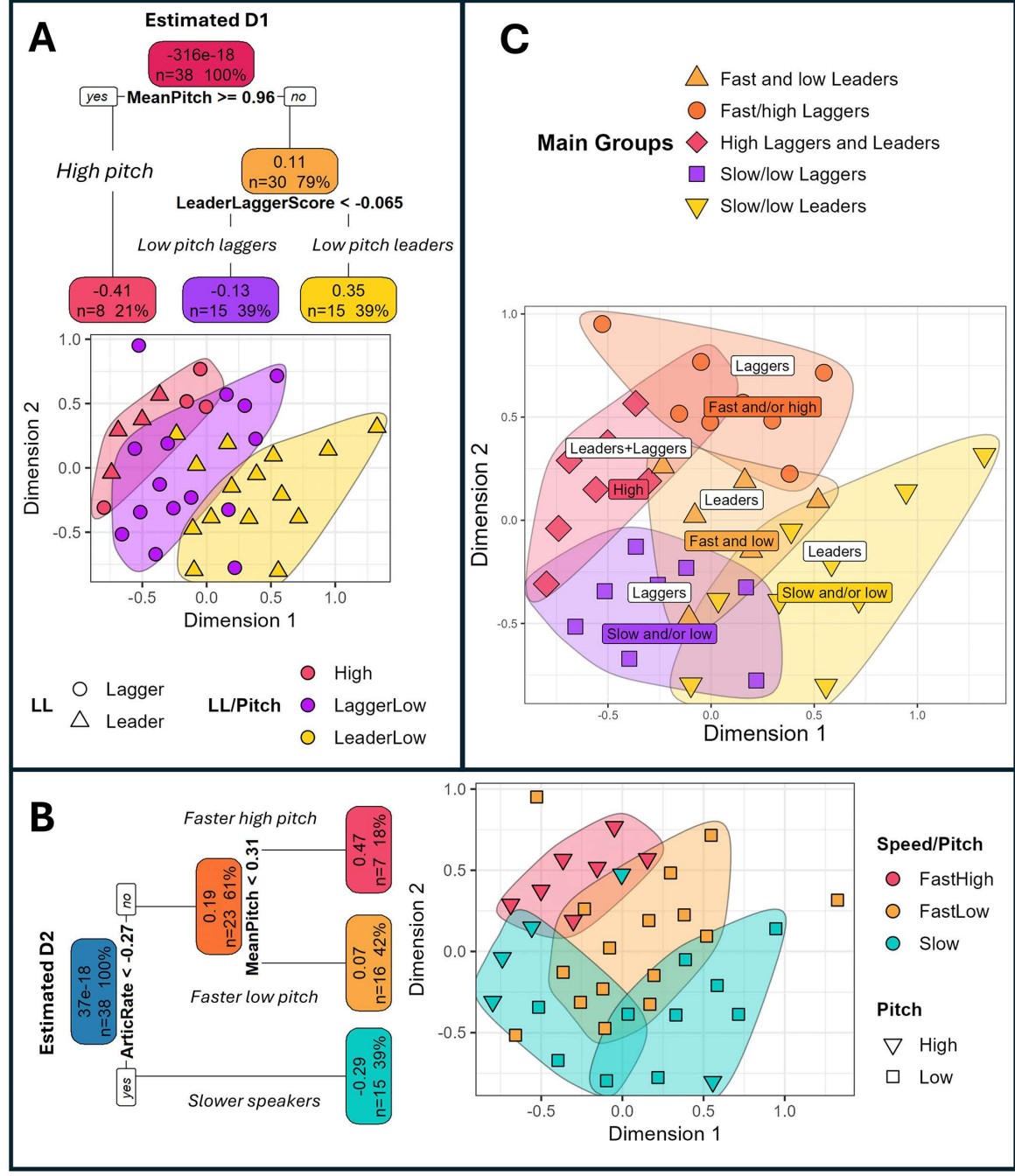

**Fig 2. (A) The results of the regression tree for Dimension 1, (B) The results of the regression tree for Dimension 2, and (C) the interpretation of the perceptual space based on (A) and (B).**

If we combine the cutoffs for speed, mean pitch, and the leader-lagger continuum, we can identify five main groups of speakers as shown in Fig 2C. The first two groups are the slower and/or lower pitched leaders (yellow upside-down triangles, bottom right) and laggers (purple squares, bottom left). Third, we have leaders who are both faster and lower

pitched (orange triangles, middle). Fourth, we have speakers who are higher pitched, regardless of whether they are a leader or lagger (red diamonds, top right). Finally, we have laggers specifically who are fast and/or higher pitched (dark orange circles, top). While there is some overlap between the different groups, there is nonetheless evidence for listeners making subtle perceptual distinctions between speakers based on all three variables. The MDS therefore points to speed and pitch, and one of the covarying NZE vowel patterns, underlying the perceptual relationships between speakers.

**4.1.2. Testing differences between the main groups in the MDS space.** We have proposed that five main speaker groups are differentiated within the MDS space. In this section, we test whether these groups are statistically distinct from each other by means of permutational MANOVA (PERMANOVA) [91,92]. PERMANOVA is a flexible statistical technique that compares the variation between groups to the variation within groups, based on a distance or dissimilarity matrix. We use the *adonis2* function from the vegan package [93] to test the null hypothesis that the centroids (mean middle point) of our five speaker groups are equivalent. While the function considers both group centroids and dispersion, the five speaker groups have comparable levels of dispersion (the average distance of a speaker from the centroid of their group is between 2.27–2.48 for all groups, see Section 2.4 of the supplementary materials).

Table 1 summarises the result of PERMOVA applied to the same dissimilarity matrix input to the MDS analysis. The results indicate that we can reject the null hypothesis that the centroids of the speaker groups are equivalent, and that the groups are, indeed, perceptually distinct. This raises the question, however, of whether all the proposed groups are distinct from each other. To investigate further, we apply the *pairwiseAdonis2* function [94], which builds on *adonis2* to conduct a pairwise comparison of each group. Table 2 summarises the pairwise contrasts, including original p-values and those adjusted using the Bonferroni correction. We can see that not all the groups are distinct from each other. Specifically, it is the group of fast leaders in the middle of the perceptual space that does not differ significantly from the other groups.

In other words, the position of the fast leaders in the middle of the perceptual space may be precisely because they share production features with each of the other four groups and are, consequently, not categorically distinct from them. Fast and low leaders are the intermediary group between the extremes of both leaders (i.e., they separate slow/low pitch

**Table 1. PERMANOVA Summary.**

|  | Df | SumOfSqs | R2 | F | Pr(>F) |
|---|---|---|---|---|---|
| **MainGroups** | 4 | 50.428 | 0.18785 | 1.9082 | 0.001*** |
| **Residual** | 33 | 218.022 | 0.81215 |  |  |
| **Total** | 37 | 268.451 | 1.00000 |  |  |

**Table 2. Pairwise PERMANOVA Summary.**

| Pairs | Df | SumsOfSqs | R2 | F.Model | p.value | p.adjusted sig |
|---|---|---|---|---|---|---|
| **Fast and low Leaders vs Slow/low Leaders** | 1 | 7.303155 | 1.09 | 0.08 | 0.409 | 1.00 |
| **Fast and low Leaders vs Fast/high Laggers** | 1 | 5.711700 | 0.84 | 0.07 | 0.685 | 1.00 |
| **Fast and low Leaders vs Slow/low Laggers** | 1 | 5.463664 | 0.88 | 0.07 | 0.626 | 1.00 |
| **Fast and low Leaders vs High Laggers and Leaders** | 1 | 7.890324 | 1.27 | 0.10 | 0.223 | 1.00 |
| **Slow/low Leaders vs Fast/high Laggers** | 1 | 17.844704 | 2.52 | 0.14 | **0.001** | **0.01*** |
| **Slow/low Leaders vs Slow/low Laggers** | 1 | 16.609823 | 2.50 | 0.14 | **0.001** | **0.01*** |
| **Slow/low Leaders vs High Laggers and Leaders** | 1 | 22.029870 | 3.31 | 0.19 | **0.001** | **0.01*** |
| **Fast/high Laggers vs Slow/low Laggers** | 1 | 16.373734 | 2.45 | 0.15 | **0.001** | **0.01*** |
| **Fast/high Laggers vs High Laggers and Leaders** | 1 | 11.269068 | 1.68 | 0.11 | **0.009** | 0.09 |
| **Slow/low Laggers vs High Laggers and Leaders** | 1 | 11.898980 | 1.92 | 0.13 | **0.015** | 0.15 |

leaders from high pitch leaders) and laggers (i.e., they separate slow/low pitch from fast/high pitch laggers). The production groups surrounding the fast leaders do, however, contrast with each other. Fast/high pitch laggers are distinct from slow/low pitch laggers, and slow/low pitch leaders are distinct from both high pitch speakers and fast/high pitch laggers. Finally, and importantly for us, slow/low leaders and laggers are significantly different from each other.

The results of the PERMANOVA support, overall, the proposed interpretation of the MDS space. They also provide additional nuance to our understanding of the relative distinctiveness of the different groups, indicating that four of the five proposed groups are statistically distinct from at least one of the other groups within the MDS space. The fifth group, fast and lower-pitch leaders, appears to be the "bridging" group at the centre of the MDS space which shares production features (speed, pitch or leader-lagger status) with speakers in the surrounding groups.

## 4.2. Predicting pairwise similarity ratings

Using MDS, we have explored the overall structure of perceptual similarity amongst our speakers. The application of MDS, which abstracts away from trial effects such as the order of the presentation of the speakers, raises a question of whether the same patterns would have emerged in a more direct analysis of the trial-by-trial pairwise similarity ratings from the online task. Are ratings of similarity symmetric, or are the properties of the first voice substantially relevant in influencing perceived similarity? To address this question, we fit a generalised additive mixed model (GAMM) using the *bam* function from the mgcv package [95] in R [9] which allowed us to explore the predictors of individual pairwise ratings, and their relationship to the perceptual space.

The dependent variable was the listener ratings for each pair of stimuli (i.e., speakers' perceived pairwise similarity). Ratings have been scaled for each participant. The predictor variables were the same four measures (leader-lagger score, back vowel configuration stimulus pitch and stimulus articulation rate) but with these measures for the first and second stimulus in each pair each fit separately as a tensor product smooth. Each tensor product smooth used four knots. Fitting tensor product smooths allows us to examine how the relationship between the first and second stimulus affects perceived similarity ratings. Moreover, opting for a GAMM over a linear mixed model allows us to account for the non-linear relationships between the independent variables in the tensor product smooths and the dependent variable. To control for the potential participant-specific impacts of the dependent variables on perceived similarity, we included a random smooth by participant ID. We also included a random smooth for each ordered pair (i.e., for stimulus *a* and stimulus *b*, there is a separate intercept for pair *ab* and for pair *ba*).

**4.2.1. Model results.** Table 3 summarises the model output, with statistically significant relationships between first and second stimulus articulation rate, pitch and the leader-lagger scores. The relationship between the first and second

**Table 3. Model summary predicting trial-by-trial pairwise similarity ratings.**

| Predictors | Estimate | Std. Error | t value | Pr(>|t|) |
|---|---|---|---|---|
| (Intercept) | 3.69768 | 0.01655 | 223.4 | <2e-16*** |
| **Smooth terms** | **Edf** | **Ref.df** | **F** | **p-value** |
| te(Stim1_ArticRate_scaled, Stim2_ArticRate_scaled) | 9.518 | 10.476 | 6.254 | <2e-16*** |
| te(Stim1_MeanPitch_scaled, Stim2_MeanPitch_scaled) | 8.056 | 9.224 | 6.232 | <2e-16*** |
| te(Stim1_LeaderLaggerScore_scaled, Stim2_LeaderLaggerScore_Scaled) | 4.726 | 5.151 | 6.172 | 8.37e-06*** |
| te(Stim1_BackVowelScore_Scaled, Stim2_BackVowelScore_Scaled) | 4.800 | 5.538 | 0.646 | 0.688 |
| s(orderedPairId) | 5.145e+02 | 1374.000 | 0.645 | <2e-16*** |
| s(workerId) | 9.619e-04 | 132.000 | 0.000 | 1.000 |
| Observations | 5054 | | | |
| R-sq.(adj) | 0.184 | | | |

stimuli back-vowel configuration scores does not significantly predict similarity ratings. As GAMM predictions are more easily understood from their visualisation than their model summaries, our focus will be on the information presented in Fig 3 and Fig 4. Each figure depicts a significant tensor smooth alongside the individual rated pairs, with the relevant measurement for the first stimulus on the horizontal axis and for the second stimulus on the vertical axis. The colour corresponds to the estimated similarity rating, with darker colours corresponding to a lower rating (lower perceived similarity), and lighter colours to a higher rating. Only model predictions that are statistically significantly different from the mean rating (alpha = 0.05) are plotted.

Fig 3A shows that pairs where the first and second stimulus are faster (top right) have higher similarity ratings (in yellow). Conversely, if the articulation rates are different (top left and bottom right), this leads to low similarity ratings (in blue), regardless of whether the slow or the fast speaker is presented first. This supports the interpretation that articulation

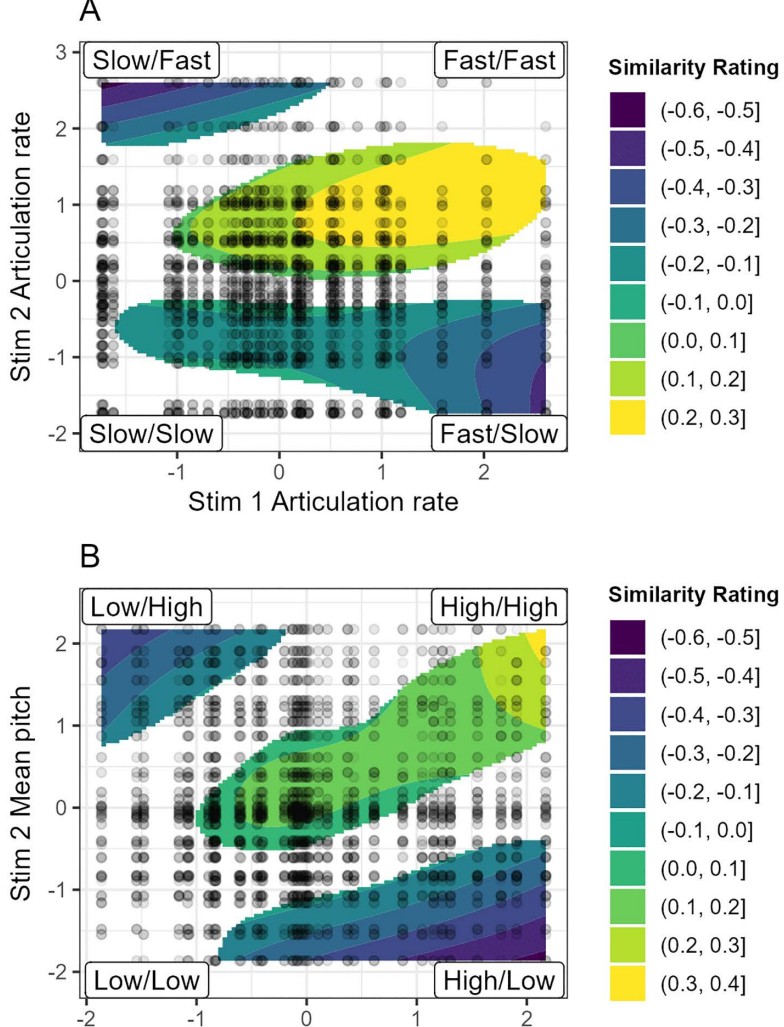

**Fig 3. Estimated similarity ratings based on first and second stimuli articulation rate (A) and pitch (B).** Darker colours correspond to lower estimated dissimilarity ratings, lighter colours to higher estimated similarity ratings.

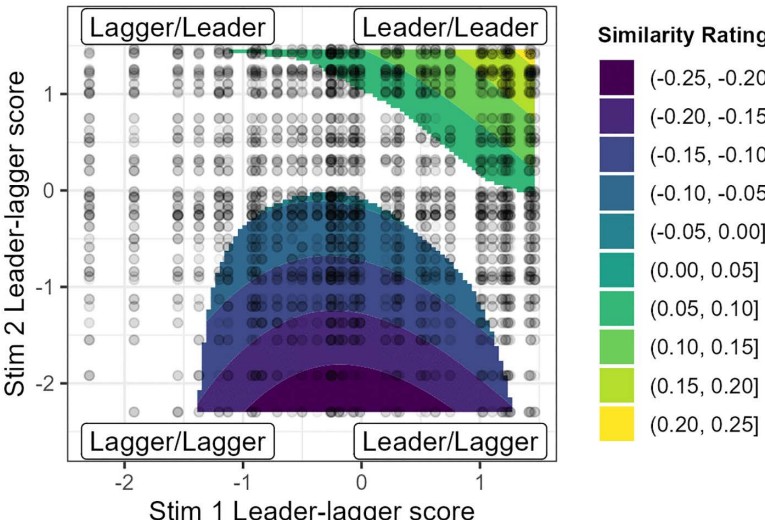

**Fig 4. Estimated similarity ratings based on leader-lagger scores.** Darker colours correspond to lower estimated dissimilarity ratings, lighter colours to higher estimated similarity ratings.

rate affects perceived similarity. Fig 3B shows a similar pattern for mean pitch, supporting the idea that pitch also affects perceived similarity, regardless of whether a high or low pitch speaker is presented first.

There is, however, a suggestion in both graphs that the effect of perceived dissimilarity may be stronger if the first speaker is faster or higher pitch (bottom right), than if the first speaker is slower or lower pitch (top left, where the blue covers a smaller area). We also note that stimuli pairs where both stimuli are slow or low pitch do not have higher pre-dicted similarity ratings (bottom left of the graphs, where we might have expected to see yellow). This suggests that higher pitch and faster speakers are more perceptually salient, leading to high similarity ratings, whereas lower pitch and slower speakers are less salient, and thus less likely to make speakers sound strikingly similar. This may lead raters to rely on other, more salient, cues to make their ratings.

Fig 4 similarly shows that pairs where both stimuli are leaders in change (top right) have higher similarity ratings. Pairs where both stimuli are laggers (bottom left) do not, suggesting that leaders are more perceptually salient than laggers. If the first speaker is a leader, then this may orient listeners to vowels, leading to high similarity when followed by a leader (top right) and low similarity if the second speaker is a lagger (bottom right). But if the first speaker is a lagger (left), this is not salient, and listeners do not use vowels to make their judgement. Another effect we see here is a general low similarity score for cases where the first speaker is not extreme in their vowels (middle of the graph), and the second speaker is also not extreme or toward the (potentially non-salient) 'lagger' end. This may suggest that speakers who are neutral on their vowels may be judged on another dimension, such as speed or pitch, which may distinguish these speakers more.

In summary, the relationship between speakers' articulation rate, pitch and leader-lagger scores all appear to have an impact on the perceived similarity of stimuli pairs, but these effects are not symmetrical. Moreover, the impact of speed and pitch looks to be both stronger (differences lead to comparatively low similarity scores, as indicated by the range of estimated similarity ratings represented in the Fig 3 and Fig 4 legends) and more consistent or symmetrical (hearing a lagger followed by a leader may not result in a lower similarity score, but hearing a faster and/or higher pitched speaker followed by a slower and/or lower pitched speaker will). The GAMM therefore provides evidence for the same variables contributing to listener evaluations as the MDS and contributes additional nuance to our understanding of how different variables predict similarity ratings.

## 5. Discussion

### 5.1. Addressing the research questions

Our primary question (RQ1) asked what structural relationships emerge among speakers when listeners evaluate their similarity based on spontaneous speech. Our second, more exploratory question, asked what factors influence pairwise similarity patterns in a trial-by-trial basis. What, then, have we learned about these questions from our two analyses?

The GAMMs indicate that articulation rate, pitch and the leader-lagger vowels affect participant trial-by-trial behaviour, while the back-vowel configuration does not. The model also provides evidence that articulation rate, pitch and leader-lagger vowels not only influence perceived similarity, but (a) their order of presentation plays a role in listener responses and (b) one end of each continuum is more perceptually salient than the other. If the first speaker is fast, high pitch, or a leader, this leads to high similarity ratings if the second speaker also shares this characteristic. However, this effect does not emerge for pairs of slow or low pitch speakers, or laggers. For the leader-lagger continuum in particular, we only see a clear effect of increased perceived similarity if the first speaker is a leader. This, together with a general effect of dissimilarity for pairs with initial 'average' speakers and following lagger/average speakers, suggest that, in the absence of hearing a leader initially, listeners may be more likely to use other characteristics to rate the speakers. Finally, the GAMM suggests that, relative to the leader-lagger vowels, the impact of speed and pitch may be both stronger and more symmetrical.

The results of the GAMM are consistent with what was learned in the MDS. Namely, leader-lagger vowels, articulation rate and pitch are relevant to how listeners perceptually differentiate between speakers. In D1, we see that high pitch speakers are perceptually distinct. Given the GAMM indicates that high pitch is more salient than low pitch, perhaps, in cases of extreme high pitch, this is the main dimension used by listeners, and the vowels are thus irrelevant. Outside of markedly high pitch speakers, however, the speaker's status as a leader or lagger becomes more important. We can also see in Fig 2C that lower pitch speakers also tend to be slower, which may facilitate the ability of listeners to orient towards vowel realisations. Turning to D2, we see a primary effect of articulation rate, mediated by pitch. Slower speakers are lower in D2, and faster speakers are higher. Combined with the evidence from the GAMM that faster articulation rate may be more salient, here we see a mediating effect at the potentially more salient end of the articulation rate continuum. It is the perception of faster speakers which is mediated by pitch, and it is fast and high pitch speakers who are particularly high on D2. That is, we see the more salient ends of continua jointly affecting perceptual similarity. This may indicate that these dimensions are working together to influence perceived similarity.

It is through combining the insights of our two analyses, then, that we get a comprehensive overall picture of the perceptual patterns. From the GAMM, we gain additional information about the speeds, pitches, and covarying vowel patterns that are likely more perceptually salient for listeners, enabling us to further interpret the patterns in the MDS. We also learn that the order stimuli are presented to listeners is relevant to individual pairwise ratings. From the MDS, we learn that pitch affects perceived similarity, mediated by the leader-lagger vowels, and articulation rate appears as a second dimension of perceived similarity, mediated by pitch. In other words, all three of these characteristics work together to determine the perceptual similarity structure, and they do not work in isolation of each other.

### 5.2. The role of covarying vowels

In this analysis we explored the degree to which two sets of co-varying vowel patterns influenced listener perception. Both the GAMM and MDS results presented above support the interpretation that one set – the 'leader-lagger' vowels – play a role in the perception of NZE voices. This supports the suggestions in Hurring et al. [17] and Brand et al. [13] that these vowels may be socially meaningful.

The second set was the configuration of a set of back vowels, which were found to covary in both Brand et al. [13] and Hurring et al. [17]. The GAMM and MDS did not provide evidence that speakers' realization of these vowels systematically

affected listener perception in our task. One potential interpretation of this lack of effect is that patterns of variation in the back vowels may not be perceptually or socio-linguistically salient and are irrelevant for listener evaluations of speaker similarity. That is, perhaps the covariation is entirely structural, and not perceptually relevant at all. Alternatively, the back vowel configuration may be perceptually or socially salient to listeners, but to a lesser extent than articulation rate, pitch and the leader-lagger vowels. Our bottom-up approach, using relatively uncontrolled stimuli, is good for picking up dominant dimensions that listeners tune into. But this does not rule out that, in more controlled stimuli, where perhaps the only thing that varied was the back vowels, that listeners may have tuned into their realization to differentiate between speakers.

### 5.3. The role of articulation rate and pitch

The combined results from the MDS and GAMM provide evidence for the leader-lagger continuum being one of multiple cues that may be used to assess speaker similarity, alongside articulation rate and mean pitch. Indeed, the perceptual salience of speed and pitch look to be greater than the leader-lagger continuum.

A question that remains is the degree to which speed and pitch are socially evaluated and truly working together with the vowels. Our results could arise from different listeners doing the task differently – some reacting to surface acoustic characteristics of the voices, others conducting the intended social evaluation. Or they could arise from different pairs of stimuli being rated on different dimensions, which the GAMM results suggest may, at least sometimes, be the case. But the results could also arise from true assessments of social meaning, where 'fast' or 'high pitch' carry social evaluations, and in a complex way that interacts with each other, and with the realization of whether a speaker is a lagger or a leader. An important topic for future work will be to conduct tasks which reveal what listeners are doing when they rate pairs of speakers. When we ask why listeners rate two high pitch speakers as similar, for example, would they tell us it is because they are both high pitch (i.e., perceptually salient)? Or because they both sound young, feminine, or some other social characteristic (i.e., socio-linguistically salient)?

Understanding the role of such prosodic factors and – particularly – the degree to which they may work with segmental factors in the creation of social meaning in New Zealand English [see 96] is another important topic for future work. In general, the salience of articulation rate and pitch is consistent with previous research documenting these as salient acoustic features to listeners [e.g., 97,98] and our results support O'Rourke and Baltazani's [99] call for a greater focus on the nascent field of 'socioprosodics'.

### 5.4. Methodological considerations

We have used two different approaches to analyse our data. As the GAMM is modelling the trial-by-trial responses, it is effective at indicating the types of properties that are being used throughout the experiment, and the potentially complex order effects. The MDS is instead effective at abstracting away from trial-by-trial effects and leveraging the collective results to provide a robust high-level picture of the perceptual organization of these speakers. MDS can also compensate, to an extent, for the uneven number of ratings across stimuli pairs by using the information that is available to situate an individual relative to all other speakers, not just those they were paired with. For example, GAMMs can only estimate how close Speaker A is to Speaker B, or A to C. They cannot utilise other ratings, such as those between B and C, to build a map of where A sits relative to both speakers.

The model results do indicate that applying MDS both reduced noise (i.e., presenting an aggregate picture of listener perception) and reduced some information in the original data (i.e., not showing potential stimuli order effects on listener perception). However, reducing the perceptual space down to two dimensions and abstracting away from order effects meant we were also able to more easily explore interactions in this space. This is because an interaction between two of our predictors (e.g., articulation rate and pitch) is now a simple two-way interaction, rather than a four-way interaction as it would have been in the GAMM (the articulation rate and pitch of each of the two stimuli). There is, therefore, clear value

in approaching perceptual similarity data from multiple angles in explorations of listener perception, and the utility and applicability of MDS will, ultimately, depend on the research question and variables of interest.

The order effects in our pairwise similarity rating task indicated by the GAMM are, nonetheless, important. Our results are consistent with work on similarity, which suggests that this is not a symmetric concept [69,100] and point to the importance of future work considering how variable/variant salience might affect participant behaviour in a rating task [64, see 65,66]. There are clear methodological implications that need to be considered, and interesting opportunities for revealing potential differences in salience. Moreover, there is work on pairwise ratings of voice similarity in the forensic literature, where the question of perceived similarity is relevant due to the use of 'voice line-ups' [see, e.g., 74,101]. While the use of MDS is recommended [e.g., 41], order effects may be important to understand when applying such work in a practical context.

## 5.5. Limitations

In this section we will discuss the main limitations of the analysis. First, the main limitation of the experiment design is that not all listeners heard all pairs. Moreover, participant dropout meant that not all pairs were heard the same number of times. On average, each pair was rated seven times (the median number of ratings is also seven), and the distribution of ratings has a standard deviation of 1.6 and variance of 2.5. It is, therefore, likely that some mean ratings used to create the similarity matrix are more informative than others. The order effects observed in the GAMM results may also be affected by a more even distribution of the pairwise ratings. Future pre-registered modelling that tests higher-order interactions with a set of stimuli that is more balanced and more evenly distributed across participants is required to fully disentangle the potential order effects at play in the perception of speaker similarity.

Second, we note that the GAMM only explained 18.4% of the variance in the data ($R^2$ measure in Table 3). It is, therefore, highly likely that factors other than those we fit to the model are relevant to the perceptual differentiation of speakers. It is also possible (some of) the variables we did include are functioning, to varying extents, as proxies for other features that more strongly underly perceived similarity. For example, articulation rate is related to, but does not capture, use of pauses and mean pitch is related to, but distinct from, other voice quality measures such as spectral tilt and shimmer. The leader-lagger and back-vowel configuration scores also capture information about two sets of individual variables which may be variably relevant to listener perception. Modelling the relative importance of different speech features to perceived speaker similarity is a clear avenue for future research.

Relatedly, the use of uncontrolled audio stimuli introduces multiple limitations. While we checked for certain forms of content (Section 3.1), we did not control for all potential lexical or topical influences on listener evaluations. More prominently, the distribution of vowels from the two clusters is not even across stimuli. There are fewer back vowels than leader-lagger vowels, providing less evidence of their realization in the stimuli that we played. As such, a third possible explanation for the lack of effect of the back-vowel configuration is that listeners were simply not exposed to sufficient tokens to make judgements based on their realisations. The back-vowel configuration may, therefore, be perceptually or socially salient, but we would have needed more targeted recordings with evenly distributed vowel tokens to reveal it.

Finally, we would like to discuss statistical power. We did not conduct a power analysis to determine the desired participant sample size because dimension-reduction techniques such as our preregistered MDS analysis do not test a null hypothesis. MDS does not, by extension, produce the Type I or Type II statistical errors power analyses are intended to mitigate [102]. The extent to which larger listener sample sizes improve the fit (i.e., stress) of an MDS analysis is, nonetheless, a relevant and under-explored question. Rodgers [103] found samples as small as one to six participants can provide good Metric Recovery of original distances and comparable stress values to larger samples, and our sample size exceeds both those numbers and common participant numbers in applications of MDS in psychology [see 104] and linguistics [e.g., 78,80,105]. Discussions of stress and "sample size" in applications of MDS otherwise focus on the number of input items [e.g., 104,106], which in our case was constrained by QuakeBox participant demographics. The exact relationship between our participant sample size and MDS fit remains, however, an open question.

As the question of pairwise ratings emerged as our analysis progressed, we did not preregister the reported GAMM analysis or conduct a post-hoc power analysis of the participant sample size. The general risk of post-hoc power analyses aside [see 107,108], there is limited precedent for determining participant sample sizes in the application of GAMMs. Linguistic papers that discuss GAMMs and statistical power consider power across model types or in the model-fitting process [e.g., 109,110], rather than in relation to sample sizes and Type I/II error probability. Furthermore, simulation methods have been developed for assessing statistical power for generalised linear mixed models [e.g., 111,112], but they do not currently apply to the outputs of GAMM models in R. In other words, there is not yet an accessible, conventionalised, approach to informing a desired sample sizes for fitting GAMMs in linguistics. As such, we highlight simulation-based approaches to participant sampling in both MDS and GAMMs as potential directions of future research and methodological innovation.

## 6. Conclusion

Past work has identified two sets of vowels in New Zealand English which work together as vowel subsystems in production. We were interested in the degree to which variation within these subsystems played a role in listener perception. To test this, we ran a pairwise-similarity task with New Zealand voices and investigated whether differences in covarying vowel patterns affected perceived speaker similarity (i.e., are perceptually salient). We also investigated the role of mean pitch and articulation rate.

Across two analyses, we found evidence that one of the vowel subsystems is perceptually salient – a set of vowels which are undergoing change in New Zealand English, and for which speakers tend to be 'laggers' or 'leaders' in the change. This did not affect similarity in isolation though. Both mean pitch and articulation rate also played a role and – indeed – these were somewhat more predictive of similarity ratings than the vowel realizations. Our analysis also suggests that perceived similarity is not symmetric, and that the degree of salience of a feature differs at different ends of the continuum. If two speakers are leaders in the sound changes, for example, this makes them sound more similar to each other than two speaker who are laggers.

Using a bottom-up approach, we were able to identify relationships between patterns of production and listeners' perceptions of voices, without artificially or prematurely imposing the social meanings that might be carried by these characteristics. The methodology of starting with pairwise similarity ratings is an effective strategy for revealing patterns in the way in which speakers are perceived. The resultant evidence for the perceptual salience of the leader-lagger vowels, mean pitch, and articulation rate now provides a solid foundation for future work investigating precisely how these variables work together in the creation of social meaning in New Zealand English

## Acknowledgments

We would like to acknowledge the Speech Communication Research Group at Northwestern and Chun-Liang Chan for the original development of the software underpinning our experiment. We thank Robert Fromont and Wakayo Mattingley for their technical support implementing the online experiment, and Gia Hurring and our colleagues at the New Zealand Institute of Language, Brain and Behaviour who have followed this project since its inception. Finally, we would like to thank all the listeners who participated in our experiment and the speakers who contributed their voices to the Quakebox.

## Author contributions

**Conceptualization:** Elena Sheard, Jen Hay, Joshua Wilson Black, Lynn Clark.

**Formal analysis:** Elena Sheard, Joshua Wilson Black.

**Funding acquisition:** Jen Hay, Lynn Clark.

**Investigation:** Elena Sheard.

**Methodology:** Elena Sheard, Jen Hay, Joshua Wilson Black.

**Software:** Joshua Wilson Black.

**Visualization:** Elena Sheard.

**Writing – original draft:** Elena Sheard, Jen Hay, Lynn Clark.

**Writing – review & editing:** Elena Sheard, Jen Hay, Joshua Wilson Black, Lynn Clark.

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
