## [Decision Letter · Decision Letter 0]

21 Jun 2025

PONE-D-25-18701Do 'leaders' in change sound different from 'laggers’? The perceptual similarity of New Zealand English voicesPLOS ONE?

Dear Dr. Sheard,

Thank you for submitting your manuscript to PLOS ONE. After careful consideration, we feel that it has merit but does not fully meet PLOS ONE’s publication criteria as it currently stands. Therefore, we invite you to submit a revised version of the manuscript that addresses the points raised during the review process.

We look forward to receiving your revised manuscript.

Kind regards,

Anirban Bhowmick, Ph.D.

Academic Editor

PLOS ONE

Journal Requirements:

This research was supported by a Royal Society of New Zealand Marsden Research Grant (21-UOC-107).

Reviewers' comments:

Reviewer's Responses to Questions

**Comments to the Author**

1. Is the manuscript technically sound, and do the data support the conclusions?

Reviewer #1: Partly

Reviewer #2: Yes

2. Has the statistical analysis been performed appropriately and rigorously?

Reviewer #1: No

Reviewer #2: Yes

3. Have the authors made all data underlying the findings in their manuscript fully available?

Reviewer #1: Yes

Reviewer #2: Yes

4. Is the manuscript presented in an intelligible fashion and written in standard English?

Reviewer #1: Yes

Reviewer #2: Yes

Reviewer #1: The manuscript presents an important contribution to sociophonetics and perceptual sociolinguistics. It offers a novel approach to understanding how listeners differentiate between speakers based on ongoing phonetic change in New Zealand English. However I have certain questions in my mind, which are as follows:

(i) Why was a two-dimensional solution chosen for the MDS analysis without reporting standard diagnostics such as stress values, a scree plot, or Shepard diagram?

(ii) Given that the GAMM explains only 18.4% of the variance, have the authors considered including additional predictors such as voice quality features (e.g., H1-H2, jitter, shimmer) or lexical/semantic content to improve model fit?

(iii) The term "markedness" is central to the discussion of pitch and articulation rate—how is this concept operationalized or validated in the study?

(iv) Have the authors conducted formal statistical tests (e.g., permutation MANOVA or cluster validity indices) to assess whether the perceptual clusters identified via MDS (e.g., leaders vs. laggers) are statistically distinct?

(v) How evenly were the 703 speaker pairs distributed among participants, and how might any imbalance have affected the construction of the perceptual similarity matrix?

(vi) Given that the final sample size (n = 133) falls short of the preregistered goal (n = 180–200), has a post hoc power analysis been conducted to assess the impact of reduced sample size on the reliability of findings?

Reviewer #2: The manuscript titled "Do 'leaders' in change sound different from 'laggers’? The perceptual similarity of New Zealand English voices" presents a perceptual sociophonetic study exploring whether listeners perceive systematic differences between speakers positioned as 'leaders' or 'laggers' in ongoing vowel changes in New Zealand English. Through a well-designed pairwise similarity rating task using spontaneous speech from a controlled speaker group, the authors use Multi-Dimensional Scaling (MDS) and Generalized Additive Mixed Models (GAMMs) to investigate the perceptual salience of vowel systems, articulation rate, and pitch. The study contributes both methodologically and theoretically to sociophonetics and sociolinguistic perception research.

Strengths

Innovative Design: The use of spontaneous speech in a pairwise similarity task is commendable, avoiding the artificiality of matched guise techniques and reflecting natural listener judgments.

Robust Statistical Analysis: The combination of MDS and GAMM offers a multifaceted understanding of how pitch, rate, and vowel changes contribute to perceptual similarity.

Clear Theoretical Framing: The manuscript builds well on prior work in NZE variation (e.g., Brand et al., Hurring et al.) and links production data to perceptual salience.

Transparency and Open Science: Pre-registration and full availability of data and code via GitHub enhance replicability and credibility.

Clarity and Structure: The writing is well-structured, with careful explanation of technical methods, aiding interdisciplinary comprehension.

Weaknesses and Suggestions for Improvement

Back-Vowel Configuration Interpretation: The null finding for back-vowel configuration is acknowledged but remains somewhat under-explored. The authors could enrich the discussion by more explicitly considering how perceptual salience may differ across vowel classes or whether methodological limitations (e.g., short stimuli) may have obscured perceptual effects.

Social Meaning Attribution: While the paper remains agnostic about whether pitch and rate are socially evaluated versus acoustically salient, a clearer separation or operationalization of these constructs (perhaps in future work) would improve clarity.

Sample Size and Power: The authors mention falling short of the preregistered participant target. Although analyses are still valid, a brief power analysis (even retrospective) would help assure readers of the robustness of perceptual inferences.

Stimulus Contextual Control: Although spontaneous speech enhances ecological validity, were there any checks for potential lexical or topical influences (e.g., emotionally marked words) that might bias perception?

Figure Interpretability: Figures 2C and 3–4 could benefit from more intuitive legends or speaker clustering highlights to help visually align MDS output with theoretical interpretations (leaders/laggers, pitch, rate).

Minor Points

Some citations in the literature review could be updated with more recent perceptual sociophonetic studies from other English varieties or multilingual contexts for comparative value.

Ensure consistent terminology (e.g., “speaker pitch” vs. “mean pitch”) throughout for clarity.

Recommendation

Accept with Minor Revisions

This is a high-quality, thoughtfully executed manuscript that significantly contributes to sociophonetic perception research. I recommend minor revisions focused on strengthening discussion points and clarifying visuals rather than methodological changes.

**Do you want your identity to be public for this peer review?** For information about this choice, including consent withdrawal, please see our Privacy Policy

Reviewer #1: No

Reviewer #2: **Yes: ** Dr. Urooj Fatima Alvi, Assistant Prpfessor, Applied Linguistics urooj.alvi@ue.edu.pk

---

## [Author Response · Author response to Decision Letter 1]

1 Aug 2025

We sincerely thank the reviewers for their time and effort in reviewing this paper. The following lists our responses to comments and questions. We also refer the editor to the attached word doc, which highlights our responses in bold.

Reviewer #1:

The manuscript presents an important contribution to sociophonetics and perceptual sociolinguistics. It offers a novel approach to understanding how listeners differentiate between speakers based on ongoing phonetic change in New Zealand English.

However I have certain questions in my mind, which are as follows:

(i) Why was a two-dimensional solution chosen for the MDS analysis without reporting standard diagnostics such as stress values, a scree plot, or Shepard diagram?

The process by which we selected a two-dimensional MDS was outlined in detail in the supplementary materials. We now also report this information in the main text in section 4.1, including stress for the reported MDS. Further details remain available in the supplementary materials, which now include scree, Shepard and bubble plots along with an additional test for insufficient dimensions using bootstrapping and permutation.

(ii) Given that the GAMM explains only 18.4% of the variance, have the authors considered including additional predictors such as voice quality features (e.g., H1-H2, jitter, shimmer) or lexical/semantic content to improve model fit?

Our research question was whether the same predictors we tested on the MDS space also predicted the pairwise similarity ratings. The available evidence is that they do, but that other predictors very likely also a play. For this question we have taken the approach of fitting models with only the directly relevant predictors. Rather than trying to fit another model or expand on this one, especially given the very extensive list of features that could be contributing to perceived speaker similarity, we have directly acknowledged the low explained variance of the reported model and its likely implications in the new limitations section and thank the reviewer for raising this (5.5).

(iii) The term "markedness" is central to the discussion of pitch and articulation rate—how is this concept operationalized or validated in the study?

We have here used ‘markedness’ interchangeably with salience. Rather than use two terms which both lack a consensus definition in linguistics, we now only use ‘salience/salient’ in the manuscript. We also clarify the two components of salience as used in the context of this analysis. One is sociolinguistic salience, which refers to the greater impact of non-standard/innovative variants on listener evaluations of speakers (i.e., listener awareness of variation and corresponding social meanings). The other is perceptual salience (i.e., the noticeability of one feature relative to another).

(iv) Have the authors conducted formal statistical tests (e.g., permutation MANOVA or cluster validity indices) to assess whether the perceptual clusters identified via MDS (e.g., leaders vs. laggers) are statistically distinct?

Thank you for raising this, we had not conducted any formal statistical tests of the clusters themselves. In response to this suggestion, we have applied permutational MANOVA and report its results in a new subsection of the results (Section 4.1.2). The PERMANOVA adds some nuance to our interpretation of the MDS and ultimately upholds the main conclusions.

(v) How evenly were the 703 speaker pairs distributed among participants, and how might any imbalance have affected the construction of the perceptual similarity matrix?

We now report the mean (7 ratings), median (7 ratings), SD (1.6) and variance (2.6) of the distribution of pair ratings in the Limitations section and supplementary materials. It is possible that certain means in the perceptual similarity matrix reflect ‘truer’ means (i.e., come from more speakers) than others, and this is now noted in the limitations section.

(vi) Given that the final sample size (n = 133) falls short of the preregistered goal (n = 180–200), has a post hoc power analysis been conducted to assess the impact of reduced sample size on the reliability of findings?

As the question of pairwise ratings emerged as our analysis progressed, we only pre-registered the MDS analysis. To our knowledge, there is not a conventional power analysis for determining the requisite participant sample for MDS, especially MDS fit to a single matrix of mean similarity measures (necessitated here by participants not rating all possible pairs). For instance, none of the MDS studies we cite mention power analysis for selection of sample sizes. The Stress value reflects the goodness-of-fit of the analysis, and we have introduced a novel, publicly available, methodology for assessing the extent to which adding in further dimensions improves the fit. We now explicitly note the novelty of this methodology in Section 4.1 and report the stress value of the reported MDS.

We have also not conducted a post-hoc power analysis for the GAMM. The risks of post-hoc power analyses aside, as far as we are aware, the available packages for calculating statistical power in R are not applicable to GAMM outputs. The linguistic papers on GAMMs that discuss statistical power focus on power across different model types (e.g., Baayen et al 2017) or in the GAMM fitting process (e.g., Soskuthy 2021), rather than power in relation to sample size selection. More complex simulation methods exist for assessing statistical power in generalised linear models (e.g., the sim.glmm function, glmmrBase package) that could be extended to GAMMs. However, developing such a method is beyond the scope of this analysis. We have stated explicitly in the paper that the second analysis is exploratory and acknowledge the potential issues of the sample distribution in the new limitations section.

However, if there are established means of calculating the effect of sample sizes on the reliability of non-individual MDS analysis and tensor smooths in GAMMs that we have missed, we are willing to apply such tests and would appreciate further details on available R packages for doing so.

Reviewer #2:

The manuscript titled "Do 'leaders' in change sound different from 'laggers’? The perceptual similarity of New Zealand English voices" presents a perceptual sociophonetic study exploring whether listeners perceive systematic differences between speakers positioned as 'leaders' or 'laggers' in ongoing vowel changes in New Zealand English. Through a well-designed pairwise similarity rating task using spontaneous speech from a controlled speaker group, the authors use Multi-Dimensional Scaling (MDS) and Generalized Additive Mixed Models (GAMMs) to investigate the perceptual salience of vowel systems, articulation rate, and pitch. The study contributes both methodologically and theoretically to sociophonetics and sociolinguistic perception research.

Weaknesses and Suggestions for Improvement

Back-Vowel Configuration Interpretation: The null finding for back-vowel configuration is acknowledged but remains somewhat under-explored. The authors could enrich the discussion by more explicitly considering how perceptual salience may differ across vowel classes or whether methodological limitations (e.g., short stimuli) may have obscured perceptual effects.

In response to this feedback, we discuss the potential impacts of the choice of stimuli on the null result for the back vowels in the new limitations section (Section 5.5). We also note the potential differences in salience between the different vowels in Section 5.2.

Social Meaning Attribution: While the paper remains agnostic about whether pitch and rate are socially evaluated versus acoustically salient, a clearer separation or operationalization of these constructs (perhaps in future work) would improve clarity.

In response to this feedback, we have differentiated between perceptual/acoustic salience and sociolinguistic salience (Section 2.3) and have been more explicit about which is being referred to in the discussion section.

Sample Size and Power: The authors mention falling short of the preregistered participant target. Although analyses are still valid, a brief power analysis (even retrospective) would help assure readers of the robustness of perceptual inferences.

We acknowledge that some form of power analysis would be ideal. However, there are not established or straightforward means of applying power analysis (in terms of participant numbers) for either MDS or GAMMs. We refer the reviewer to our response to reviewer 1 for more details.

Stimulus Contextual Control: Although spontaneous speech enhances ecological validity, were there any checks for potential lexical or topical influences (e.g., emotionally marked words) that might bias perception?

We state in the supplementary materials that stimuli were checked for lexical items that would be potential markers of socioeconomic status, and that did not contain explicitly negative content about their experiences of the Christchurch earthquake. This information has now also been included in the main text. We also state in the limitations section that there may be different lexical affects beyond those we checked for.

Figure Interpretability: Figures 2C and 3–4 could benefit from more intuitive legends or speaker clustering highlights to help visually align MDS output with theoretical interpretations (leaders/laggers, pitch, rate).

In response to this feedback, we have updated Figures 3 and 4 with labels to assist in interpretability. We have also included more details in the figure captions to explain the legend colours. We are happy to make further changes to Figure 2C but were not able to identify exactly what was causing confusion in the original version.

Minor Points

Some citations in the literature review could be updated with more recent perceptual sociophonetic studies from other English varieties or multilingual contexts for comparative value.

In response to this feedback, we have updated/added to some citations in the literature review with recent perceptual studies of other English and non-English varieties, and second language acquisition contexts

Ensure consistent terminology (e.g., “speaker pitch” vs. “mean pitch”) throughout for clarity.

We have made adjustments to the terminology throughout.

---

## [Decision Letter · Decision Letter 1]

18 Sep 2025

PONE-D-25-18701R1Do 'leaders' in change sound different from 'laggers’? The perceptual similarity of New Zealand English voicesPLOS ONE?

Dear Dr. Sheard,

Thank you for submitting your manuscript to PLOS ONE. After careful consideration, we feel that it has merit but does not fully meet PLOS ONE’s publication criteria as it currently stands. Therefore, we invite you to submit a revised version of the manuscript that addresses the points raised during the review process.

We look forward to receiving your revised manuscript.

Kind regards,

Anirban Bhowmick, Ph.D.

Academic Editor

PLOS ONE

Journal Requirements:

Reviewers' comments:

Reviewer's Responses to Questions

**Comments to the Author**

Reviewer #1: All comments have been addressed

Reviewer #3: All comments have been addressed

2. Is the manuscript technically sound, and do the data support the conclusions?

Reviewer #1: Yes

Reviewer #3: Yes

3. Has the statistical analysis been performed appropriately and rigorously?

Reviewer #1: Yes

Reviewer #3: Yes

4. Have the authors made all data underlying the findings in their manuscript fully available?

Reviewer #1: Yes

Reviewer #3: Yes

5. Is the manuscript presented in an intelligible fashion and written in standard English?

Reviewer #1: Yes

Reviewer #3: Yes

Reviewer #1: The authors have addressed all substantive reviewer concerns and significantly strengthened the paper. With minor refinements to figures and a forward-looking discussion of sample size/power issues, this manuscript will be ready for publication.

Reviewer #3: I read the ms a few times. The authors did a great job revising the manuscript. The paper's structure became clearer, which improved its readability. Also, the methodological choices are well explained. I have no further questions.

**Do you want your identity to be public for this peer review?** For information about this choice, including consent withdrawal, please see our Privacy Policy

Reviewer #1: No

Reviewer #3: **Yes: ** Masako Hirotani

---

## [Author Response · Author response to Decision Letter 2]

13 Oct 2025

We sincerely thank the reviewers for their time spent reviewing the revised manuscript. The following document lists all comments and feedback, with our responses in bold.

Comments to the Author

1. If the authors have adequately addressed your comments raised in a previous round of review and you feel that this manuscript is now acceptable for publication, you may indicate that here to bypass the “Comments to the Author” section, enter your conflict of interest statement in the “Confidential to Editor” section, and submit your "Accept" recommendation.

Reviewer #1: All comments have been addressed

Reviewer #3: All comments have been addressed

2. Is the manuscript technically sound, and do the data support the conclusions?

Reviewer #1: Yes

Reviewer #3: Yes

3. Has the statistical analysis been performed appropriately and rigorously?

Reviewer #1: Yes

Reviewer #3: Yes

4. Have the authors made all data underlying the findings in their manuscript fully available?

Reviewer #1: Yes

Reviewer #3: Yes

5. Is the manuscript presented in an intelligible fashion and written in standard English?

Reviewer #1: Yes

Reviewer #3: Yes

6. Review Comments to the Author

Reviewer #1: The authors have addressed all substantive reviewer concerns and significantly strengthened the paper. With minor refinements to figures and a forward-looking discussion of sample size/power issues, this manuscript will be ready for publication.

In response to this feedback, we have integrated a discussion of statistical power and sample sizes into our ‘Limitations’ section of the manuscript. We specifically state:

“Finally, we would like to discuss statistical power. We did not conduct a power analysis to determine the desired participant sample size because dimension-reduction techniques such as our pre-registered MDS analysis do not test a null hypothesis. MDS does not, by extension, produce the Type I or Type II statistical errors power analyses are intended to mitigate [103]. The extent to which larger listener sample sizes improve the fit (i.e., stress) of an MDS analysis is, nonetheless, a relevant and underexplored question. Rodgers [106] found samples as small as one to six participants can provide good Metrix Recovery of original distances and comparable stress values to larger samples, and our sample size exceeds both those numbers and common participant numbers applications of MDS in psychology [see 104] and linguistics [e.g., 78, 80, 105]. Discussions of stress and “sample size” in applications of MDS otherwise focus on the number of input items [e.g., 104, 107], which in our case was constrained by QuakeBox participant demographics. The exact relationship between our participant sample size and MDS fit remains, however, an open question.

As the question of pairwise ratings emerged as our analysis progressed, we did not pre-register the reported GAMM analysis or conduct a post-hoc power analysis of the participant sample size. The general risk of post-hoc power analyses aside [see 108, 109], there is limited precedent for determining participant sample sizes in the application of GAMMs. Linguistic papers that discuss GAMMs and statistical power consider power across model types or in the model-fitting process [e.g., 110, 111], rather than in relation to sample sizes and Type I/II error probability. Furthermore, simulation methods have been developed for assessing statistical power for generalised linear mixed models [e.g., 112, 113], but they do not currently apply to the outputs of GAMM models in R. In other words, there is not yet an accessible, conventionalised, approach to informing a desired sample sizes for fitting GAMMs in linguistics. As such, we highlight simulation-based approaches to participant sampling in both MDS and GAMMs as potential directions of future research and methodological innovation.”

We have re-generated the figures to align with PLOS One’s requirements and run them through the required PACE tool. We have also adjusted Figure 2 to be more readable in terms of font and legend sizes and updated the formatting of the manuscript text and tables.

Reviewer #3: I read the ms a few times. The authors did a great job revising the manuscript. The paper's structure became clearer, which improved its readability. Also, the methodological choices are well explained. I have no further questions.

We thank the reviewer for their kind comments.

---

## [Decision Letter · Decision Letter 2]

19 Nov 2025

Do 'leaders' in change sound different from 'laggers’? The perceptual similarity of New Zealand English voices

PONE-D-25-18701R2

Dear Dr. Sheard,

We’re pleased to inform you that your manuscript has been judged scientifically suitable for publication and will be formally accepted for publication once it meets all outstanding technical requirements.

Kind regards,

Anirban Bhowmick, Ph.D.

Academic Editor

PLOS ONE

Additional Editor Comments (optional):

Reviewers' comments:

Reviewer's Responses to Questions

**Comments to the Author**

Reviewer #1: All comments have been addressed

Reviewer #3: All comments have been addressed

2. Is the manuscript technically sound, and do the data support the conclusions?

Reviewer #1: Yes

Reviewer #3: Yes

3. Has the statistical analysis been performed appropriately and rigorously?

Reviewer #1: Yes

Reviewer #3: Yes

4. Have the authors made all data underlying the findings in their manuscript fully available?

Reviewer #1: Yes

Reviewer #3: Yes

5. Is the manuscript presented in an intelligible fashion and written in standard English?

Reviewer #1: Yes

Reviewer #3: Yes

Reviewer #1: The authors have addressed the questions; the paper structure became clearer and improved the readability. Hence, the paper is ready for publication.

Reviewer #3: The authors did a good job addressing the reviewers' comments, and the work is certainly interesting.

**Do you want your identity to be public for this peer review?** For information about this choice, including consent withdrawal, please see our Privacy Policy

Reviewer #1: No

Reviewer #3: **Yes: ** Masako Hirotani

---

## [Editor Report · Acceptance letter]

PONE-D-25-18701R2

PLOS ONE

Dear Dr. Sheard,

I'm pleased to inform you that your manuscript has been deemed suitable for publication in PLOS ONE. Congratulations! Your manuscript is now being handed over to our production team.

Kind regards,

on behalf of

Dr. Anirban Bhowmick

Academic Editor

PLOS ONE